# The Geometry of Representational Failures in Vision Language Models

**Daniele Savietto** [1]  **Declan Campbell** [2]  **André Panisson** [3]  **Marco Nurisso** [4]
**Giovanni Petri** [5]  **Jonathan D. Cohen** [2]  **Alan Perotti** [3]

## Abstract

Vision-Language Models (VLMs) exhibit puzzling failures in multi-object visual tasks, such as hallucinating non-existent elements or failing to identify the most similar objects among distractions. While these errors mirror human cognitive constraints, such as the "Binding Problem", the internal mechanisms driving them in artificial systems remain poorly understood. Here, we propose a mechanistic insight by analyzing the representational geometry of open-weight VLMs (Qwen, InternVL, Gemma), comparing methodologies to distill "concept vectors" – latent directions encoding visual concepts. We validate our concept vectors via steering interventions that reliably manipulate model behavior in both simplified and naturalistic vision tasks (e.g., forcing the model to perceive a red flower as blue). We observe that the geometric overlap between these vectors strongly correlates with specific error patterns, offering a grounded quantitative framework to understand how internal representations shape model behavior and drive visual failures.

## 1. Introduction

Vision-Language Models (VLMs) have achieved remarkable capabilities in describing and reasoning about complex visual scenes, yet they exhibit puzzling failures on seemingly simple multi-object tasks: miscounting objects, confusing which color belongs to which shape, and producing "illusory conjunctions" that mirror errors observed in human rapid visual processing (Campbell et al., 2024; Rahmanzadeh-hgervi et al., 2024). These failures point to fundamental limitations in how VLMs process multi-object scenes that, in turn, may reflect important underlying principles of function. However, the internal drivers of these errors remain poorly understood.

We propose that these limitations are best understood as a general problem of **geometric representational interference**, where the high-dimensional vectors encoding distinct concepts clash within a shared latent space. This geometric interference mechanism serves as a unifying framework for diverse failure modes: it parallels the principles of representational interference used to model working memory (Oberauer & Lin, 2017), where close feature values in a continuous space can clash, and it manifests as the classical "binding problem" when shape and color features become entangled (Treisman & Gelade, 1980; Von Der Malsburg, 1994). In biological vision, serial attention mitigates this interference by isolating objects in time. VLMs, however, lack this temporal dimension and must thus resolve this task in a single feedforward pass. Consequently, they often fail to maintain orthogonal representations for co-occurring objects, leading to unavoidable feature blending in high-interference scenarios (e.g., seeing a red circle when the stimulus actually contains a red square and a green circle).

While recent work has documented these failures in proprietary VLMs at the behavioral level (Campbell et al., 2024), the representational geometry driving them is still not well understood. This leaves open critical questions:

1. Do these errors reflect systematic constraints arising from the architectural design and training paradigm of current VLMs, or artifacts of specific model conditions?
2. Can signatures of the geometric interference responsible for both binding failures and compression artifacts be detected directly in the latent space?
3. Can mechanistic interpretability methods reveal such structure and the extent to which it causally determines model behavior?

Here, we address these questions in open-weight VLMs, combining behavioral experiments, representational analysis, and causal validation. Our findings support three central claims:

---

[1]Dipartimento di Fisica, Università di Torino [2]Princeton Neuroscience Institute and AI Lab, Princeton University [3]Intesa Sanpaolo AI Research [4]Dipartimento di Scienze Matematiche, Politecnico di Torino [5]NPLab, Network Science Institute, Northeastern University London, London, UK. Correspondence to: Daniele Savietto <daniele.savietto@edu.unito.it>.

*Proceedings of the $43^{rd}$ International Conference on Machine Learning*, Seoul, South Korea. PMLR 306, 2026. Copyright 2026 by the author(s).

- **VLMs develop compositional and semantically structured representations.** We derive concept vectors for visual features (colors, shapes, and their conjunctions) and show that these vectors organize into structured manifolds whose geometry mirrors semantic relationships: nearby hues cluster together, and color-shape conjunctions factorize cleanly along independent axes.
- **These representations are causally active.** We validate concept vectors via activation steering: linearly substituting one concept vector for another reliably redirects model perception – for instance, forcing the model to describe a red flower as blue – demonstrating that the extracted directions are not mere correlational artifacts but functional components of the model's computation.
- **Representational geometry correlates with downstream failures.** We show that the cosine similarity between concept vectors accounts for systematic error patterns in two multi-object tasks: visual search and color similarity estimation. Greater geometric overlap between representations consistently tracks higher error rates or lower model confidence, offering a quantitative signature of when failures are likely to occur.

Together, these results are naturally interpreted through the lens of the "Curse of Generalization" (Frankland et al., 2026): structured representations that enable flexible generalization – such as associating the label "red" to a continuous spectrum of shades across varied objects – introduce susceptibility to inter-concept interference. Crucially, this tradeoff is particularly acute for *parallel* processing systems: biological vision mitigates it through serial attention, isolating objects in time, whereas VLMs must resolve all concepts simultaneously in a single feedforward pass, leaving geometric crowding unmitigated.

The paper is structured as follows. After reviewing related work in Section 2, Section 3 introduces the methods employed to extract vector representations and the steering operation used to validate them. Section 4 applies these methods to extract and validate representations of a set of colors and a set of color-shape conjunctions. In Section 5, we investigate the compositional and semantic structure of these representations, revealing parallels with generalization gradients studied in cognitive science. Finally, Section 6 examines correlations between these representations and model outputs on specific tasks, and Section 7 draws our conclusions.

Code to reproduce the results is available at this link.

## 2. Related Work

**Modern Vision-Language Architectures.** The landscape of multimodal AI has shifted from task-specific modular networks to general-purpose Large Multimodal Models (LMMs), typically constructed by bridging a pre-trained visual encoder (e.g., CLIP-ViT, SigLIP) with an LLM backbone via a learnable interface (Yin et al., 2024). Current state-of-the-art open-weight models – including Qwen-VL (Yang et al., 2025), InternVL (Chen et al., 2024), and Gemma (Kamath et al., 2025) – adopt a "fused" architecture where visual features are projected directly into the LLM's token embedding space. Training proceeds in two stages: pre-training to align feature spaces, followed by visual instruction tuning (Liu et al., 2023). While this unified approach enables remarkable fluency, it forces continuous visual signals into a spatially-discretized grid, creating bottlenecks where fine-grained visual details may be lost or conflated.

**Mechanistic Interpretability and Steering.** Mechanistic interpretability aims to reverse-engineer neural network computations. A foundational concept is the *Linear Representation Hypothesis*: meaningful concepts are encoded as linear directions in activation space (Mikolov et al., 2013), formalized by Kim et al. (2018) with *Concept Activation Vectors*. Techniques for identifying these directions range from supervised probes (Alain & Bengio, 2016) to Sparse Autoencoders (Bricken et al., 2023). Crucially, the field has moved from passive identification to causal intervention: activation steering manipulates model behavior by injecting vectors into the residual stream, amplifying or suppressing target concepts (Turner et al., 2024; Zou et al., 2025; Templeton et al., 2024). In the multimodal domain, steering has been applied to correct physical behaviors (Sharma et al., 2024), though most approaches rely on text-derived vectors rather than direct geometric manipulation of visual concepts.

**Geometric Signatures of Failure.** Analyzing failure modes through representational geometry has become central to understanding deep networks (Papyan et al., 2020; Mamou et al., 2020; Park et al., 2025). A key insight is that object manifolds must be sufficiently flat and orthogonal to allow linear separability; under distribution shifts or multi-object settings, this geometry degrades. Theoretical constraints like the "generalization-identification tradeoff" suggest fundamental capacity limits for resolving distinct objects in a shared space (Nurisso et al., 2026), and recent hallucination metrics have shifted accordingly from text overlap to geometric measures of uncertainty (Gautam et al., 2025; Xu et al., 2025). We build on this work by linking specific geometric properties of concept vectors to binding failures.

**Connections to Human Cognition.** VLM errors mirror limitations in human visual cognition. The difficulty in correctly associating attributes with objects is functionally identical to the "binding problem": Treisman & Gelade (1980) showed that attentional overload causes "illusory conjunctions" where features from different objects blend.

This likely reflects efficient compression: both biological and synthetic systems map continuous spectra to discrete categories to optimize bandwidth and generalization (Berlin & Kay, 1969; Zaslavsky et al., 2018; Shepard, 1958). VLMs lacking serial attention must compress multi-object scenes into a single vector sequence, and understanding their failures as the mathematical cost of this compression provides a unifying framework for analyzing machine perception errors. The within-category compression we document is also reminiscent of *categorical perception*, a long-postulated property of human cognition in which stimuli falling within the same category are perceptually compressed while those across categories are separated (Harnad, 1990; Goldstone, 1994).

# 3. Methodology

To investigate the mechanisms underlying semantic and compositional visual failures in VLMs, we propose a framework for extracting, validating, and analyzing the geometry of internal representations. We first present two different methods for concept vector extraction: (i) a *supervised approach* using attention probes, and (ii) a *centroid-based approach*. We then define a causal intervention (*steering*) approach to validate these representations.

## 3.1. Preliminaries

We consider Vision-Language Models (VLMs) with a modular architecture composed of three main components: (i) a text embedding module, which maps input strings to sequences of token embeddings; (ii) a vision encoder, typically based on a Vision Transformer (ViT), which maps an input image to a sequence of visual feature vectors; and (iii) a Large Language Model (LLM), which processes sequences of embeddings and produces a probability distribution over output tokens. The vision encoder includes a learned alignment or projection layer that maps visual features into the LLM's embedding space. When a VLM is prompted to perform tasks such as image description or visual question answering, the LLM operates over a mixed sequence of text and image-derived tokens. The comparison between the task specification (expressed in natural language) and the visual content therefore occurs within the LLM's internal representations (Liu et al., 2023).

**Formal framework.** Let $\mathcal{M}$ be a Vision-Language Model with a visual encoder $E$. Given an input image $x$, we denote by $\mathbf{H} = (\mathbf{h}_1, \ldots, \mathbf{h}_L)$ the sequence of token embeddings output by $E$, where $L$ is the sequence length, and $\mathbf{h_t} \in \mathbb{R}^d$ with $d$ the hidden dimensionality. These representations are subsequently injected into the LLM and processed within its residual stream together with text token embeddings. Since all cross-modal interaction and task-dependent computation

are mediated by the LLM, we focus our analysis on these internal representations. In particular, our objective is to identify a vector $\mathbf{v}_c \in \mathbb{R}^d$ (the *concept vector*) that encodes a specific visual concept $c$ (e.g., "red", "square", or "red square") in the space of embeddings.

Although the idea of a concept vector was first introduced by Kim et al. (2018) to refer to vectors obtained by training linear classifier probes, in this paper we will use it to describe any candidate representation of a concept regardless of how it was obtained.

## 3.2. Concept Vector Distillation

A central challenge in mechanistic interpretability is ensuring that a discovered direction $\mathbf{v}_c \in \mathbb{R}^d$ is functionally relevant to the model's internal computations. We identify these directions by comparing two distinct methods: training linear classifiers probes that separate classes and identifying concept class centroids in selected sets of activations.

**Probe-Based Method.** Our first approach identifies directions by training a linear classifier to distinguish between the presence and absence of a concept. We adopt a supervised probing approach, a simplified version of the one proposed by Tenney et al. (2019). We define an attention-based probe parameterized by a learnable direction $\mathbf{u}_c \in \mathbb{R}^d$ and scalars $b_{att}, w_{out}, b_{out}$. For a sequence of token embeddings $\mathbf{H} = (\mathbf{h}_1, ..., \mathbf{h}_L)$, the probe computes an attention score $\alpha_t$ to aggregate a context vector and predict the presence of concept $e$ via a sigmoid projection $\sigma$:

$$\alpha_t = \frac{\exp(\mathbf{h}_t^\top \mathbf{u}_c + b_{att})}{\sum_{\tau=1}^{L} \exp(\mathbf{h}_\tau^\top \mathbf{u}_c + b_{att})}$$

$$\hat{y} = \sigma\left(\left(\sum_{t=1}^{L} \alpha_t \mathbf{h}_t^\top \mathbf{u}_c\right) w_{out} + b_{out}\right) \quad (1)$$

Under the linear representation hypothesis (Park et al., 2023), the learned attention vector $\hat{\mathbf{v}}_{probe}^{(c)} = \mathbf{u}_c / \|\mathbf{u}_c\|$ serves as the candidate concept vector. However, such probes may exploit discriminative shortcuts, identifying hyperplanes that maximize separation on a specific training set but fail to capture the actual, more general representational structure.

**Centroid-Based Method.** Instead of learning a boundary, this second approach recovers the concept directly from the distribution of activations. Let $C = \{c\}$ be a set of concepts, and for each of them, let $\mathcal{F}_c$ be a set of relevant token embeddings, obtained from images containing concept $c$ in many positions and contexts. We compute the class centroid vector $\boldsymbol{\mu}_c = \mathbb{E}_{\mathbf{h} \in \mathcal{F}_c}[\mathbf{h}]$; to isolate the concept-specific signal from shared features like "objectness" or "background", we project the centroid onto the subspace orthogonal to the

global activation mean vector $\boldsymbol{\mu}_{glob}$:

$$\mathbf{v}_{raw}^{(c)} = \boldsymbol{\mu}_c - \boldsymbol{\mu}_{glob}, \quad \hat{\mathbf{v}}_{rec}^{(c)} = \frac{\mathbf{v}_{raw}^{(c)}}{\|\mathbf{v}_{raw}^{(c)}\|} \quad (2)$$

**Structural Regularization through PCA.** To bridge these methods, we introduce an intermediate method: the PCA-Probe. Our setup involves concepts defined as combinations of two categorical factors (e.g., colors and geometric shapes), each with $N$ alternatives, yielding $N^2$ concept vectors (e.g., the combinations "red square", "blue triangle", etc.). Under the assumption that these two factors are represented compositionally, these vectors lie on a structured manifold induced by the Cartesian product of the two category sets, where each categorical axis has $N-1$ degrees of freedom, for a total of $2N-2$.

We exploit this structure by applying PCA to the $N^2$ probe-based vectors and retaining only the first $2N-2$ (orthonormal) principal components, discarding directions that do not align with the assumed factorial organization. Operationally, letting $V$ be the matrix whose columns are the first $2N-2$ (orthonormal) principal components, we obtain the regularized concept vectors as $\mathbf{v}_{PCA} = VV^\top \hat{\mathbf{v}}_{probe}$, i.e., the projection of each probe vector onto this low-dimensional subspace. This forces the discriminative objective to respect the underlying categorical structure, acting as an implicit geometric regularizer that discards directions not aligned with the factorial organization of the concepts.

### 3.3. Causal Validation via Activation Steering

We evaluate the causal role of concept vectors in model behavior by defining an intervention that steers the model's perception from a *source* concept $A$ to a *target* concept $B$, as instantiated by their normalized concept vectors $\hat{\mathbf{v}}_A$ and $\hat{\mathbf{v}}_B$. Given the activation sequence $\mathbf{H}$ of an image containing an object or property $A$, we apply the following transformation to every token $\mathbf{h}_t \in \mathbf{H}$:

$$\mathbf{h}_t' = \mathbf{h}_t - (\mathbf{h}_t^\top \hat{\mathbf{v}}_A)\hat{\mathbf{v}}_A + (\mathbf{h}_t^\top \hat{\mathbf{v}}_A)\hat{\mathbf{v}}_B \quad (3)$$

This operation linearly subtracts the component of $\mathbf{h}_t$ aligned with concept $A$ and injects it along the direction associated with concept $B$. If $\hat{\mathbf{v}}_A$ and $\hat{\mathbf{v}}_B$ correspond to causally meaningful internal variables, this intervention should induce the model to behave as if object $B$ were present in place of object $A$, effectively steering $A$'s representation into $B$'s.

Many steering methods in mechanistic interpretability apply a fixed steering vector with a globally chosen scaling factor, which acts as a hyperparameter controlling intervention strength (Turner et al., 2024; Rimsky et al., 2024; Chalnev et al., 2024). In contrast, our approach modulates the intervention magnitude using the activation's own projection

onto $\mathbf{v}_A$, thereby preserving the original feature intensity. By construction, this yields a localized intervention that minimizes disruption to unrelated visual information that will not be contained in $\hat{\mathbf{v}}_B - \hat{\mathbf{v}}_A$.

In our experiments, we use the empirical success rate of this steering procedure (that is, steer $A \to B$, check if model sees $B$) as an indicator of whether the extracted vectors correspond to functionally relevant internal directions.

## 4. Representation Extraction and Validation

In this section we apply the methods described in Section 3 to extract and causally validate representations of simple visual features. We first extract concept vectors for 6 colors from synthetic images of colored shapes, validating them by steering model perception on natural images and thus demonstrating that the extracted representations generalize beyond their training distribution. We then extract vectors for color-shape conjunctions, validated via steering on synthetic multi-object images. In both settings, centroid-based extraction consistently outperforms probe-based methods in producing causally meaningful representations.

We conduct systematic experiments on three open-weight vision-language models: Qwen2.5-VL 7B (Bai et al., 2025), InternVL2.5 8B (Chen et al., 2024), and Gemma 3 12B (Kamath et al., 2025). Throughout the paper, we will refer to them as Qwen, InternVL and Gemma. All models follow the conceptual architecture described in Section 3.1, but differ in several design aspects, including the number of layers and the hidden dimensionality of the ViT and LLM components. In particular, the dimensionality $d$ of the LLM residual stream in which our concept vectors are defined is 3584 for Qwen, 4096 for InternVL, and 3840 for Gemma.

Experimental details, including probe training procedure and prompts used, are provided in Appendices B and C.

### 4.1. Causal Steering of Natural Images

To validate whether our extracted concept vectors correspond to functionally relevant internal directions rather than mere correlational artifacts, we assess their ability to causally steer the model's perception in a zero-shot setting.

**Experimental Setup.** We generate concept vectors for 6 colors (red, green, blue, yellow, orange, purple) using the methodologies described in Section 3.2. For the probe-based method, we train probes on a synthetic dataset of multi-object scenes containing various colored shapes (Figure 1a). For the centroid-based method, we extract vectors from synthetic images containing a single colored shape in varying positions (Figure 1b). All synthetic images are $448 \times 448$ pixels, mapping to a sequence of $L = 256$ token embeddings in all three models.

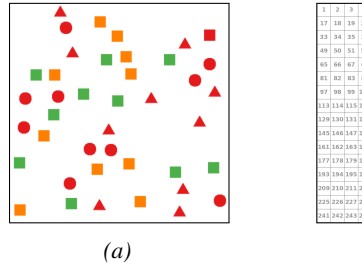 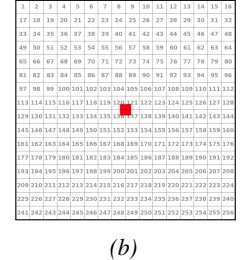

*(a)*            *(b)*

*Figure 1.* (a) Example of image used to train the attentive probes. (b) Example of image used to distill centroid-based concept vectors for "red square" – here we also show the VLM tokenization grid.

**Qualitative Validation.** While these vectors are derived from controlled synthetic environments, we first verify their transferability to natural images. Figure 2 demonstrates a qualitative success: by injecting the "blue" concept vector and subtracting "red", we force Qwen to perceive a red rose as blue while maintaining textual fluency and topical pertinence.

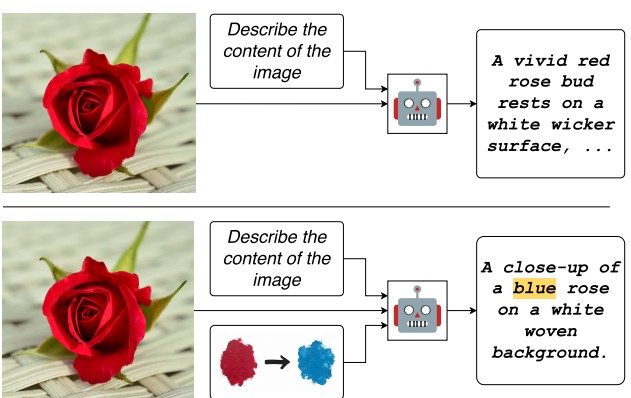

*Figure 2.* Example of causal intervention on a natural image description task. The image, prompt and model weights are unchanged; we manipulated the model's activations in order to force the 'red' color to be perceived as blue.

**Quantitative Analysis: Probes vs. Centroids.** We systematically evaluate steering performance on a custom dataset of 60 labeled real-world images (10 per color).

For each image, we perform a targeted intervention: removing the ground-truth color vector and injecting a target color vector. We prompt the model to report the color of the object represented both with and without the injection of the target color, and check how the model's responses change from the original correct color to the injected one. We measure the success rate of the model in reporting the new, injected color across 300 total steering operations (10 for each ordered color pair).

The result, summarized in Table 1, reveal a stark contrast between the two extraction methods. **(i) Failure of Naive Probing:** Raw probes (Probe) fail almost completely on Qwen and Gemma (3.7% accuracy). This supports our hypothesis that discriminative objectives, without geometric

constraints, learn "shortcut" directions that do not generalize to the model's actual structure. **(ii) Impact of Structural Regularization:** Applying PCA regularization (PCA Probe) consistently improves performance over naive probing (e.g., Qwen improves from 3.7% to 32.2%). In the case of InternVL, it even achieves peak performance (92.0%). However, this improvement is highly inconsistent across architectures, yielding only 16.3% accuracy for Gemma. **(iii) Robustness of Geometric Distillation:** The Centroid-based method demonstrates superior stability, achieving at least 84.7% accuracy across all three models.

|  | QWEN | INTERNVL | GEMMA |
|---|---|---|---|
| PROBE | 3.7% | 61.1% | 3.7% |
| PCA PROBE | 32.2% | 92.0% | 16.3% |
| CENTROID | 84.7% | 88.4% | 95.7% |

*Table 1.* Color steering accuracies on real world dataset for different VLMs and concept vector extraction methods.

Crucially, all concept vectors were distilled from synthetic data using a single RGB value per color. The fact that centroid vectors successfully steer natural images containing complex lighting and varied shades suggests they capture a robust representation of color that aligns with the model's internal concept "template".

### 4.2. Steering Compositional Concepts

To address the core of the binding problem, we verify whether VLMs represent *compositional* concepts – color-shape conjunctions like "red square" or "blue triangle" – as structured sums of their parts that can be consistently manipulated.

We extract concept vectors for 36 composite objects (6 colors × 6 shapes) using both probe-based and centroid-based methods. To validate these vectors, we perform activation steering on synthetic multi-object images. Success is defined by a strict criteria: when steering from source object $A$ to target object $B$, the model must hallucinate the target $B$ (e.g., "blue triangle") while ceasing to report the source $A$ (e.g., "red square"), without affecting a control object $C$.

Table 2 reveals a stark divergence in efficacy. First, naive probes fail entirely (0–2%), which confirms our hypothesis that unconstrained probes learn "discriminative shortcuts" – directions that separate classes in the training set but possess components orthogonal to the model's internal causal mechanism. Second, by imposing geometric constraints (PCA-Probe) or using a data-driven method (Centroid), we recover functional directions. Centroid-based vectors achieve the best performance for Qwen (78.1%) and Gemma (75.3%), suggesting that the model's internal binding mechanism is additive, aligning best with the arithmetic mean of activations.

Interestingly, InternVL shows the opposite pattern, with PCA-regularized probes (46.8%) outperforming centroids (35.9%). We note that these numbers reflect a strict three-way criterion (removal of $A$, insertion of $B$, preservation of $C$), so even moderate success rates indicate meaningful causal control.

|           | QWEN   | INTERNVL | GEMMA  |
|-----------|--------|----------|--------|
| PROBE     | 0.0%   | 2.0%     | 0.0%   |
| PCA PROBE | 44.2%  | 46.8%    | 17.9%  |
| CENTROID  | 78.1%  | 35.9%    | 75.3%  |

*Table 2.* Steering success for compositional concepts (color-shape pairs) on synthetic images. Success requires the model to report seeing the target object $B$ after steering, while no longer reporting the source object $A$.

# 5. Representational Structure of Visual Concepts

Having established that concept vectors are causally meaningful, we now examine their geometric organization. We first analyze the vectors representing color-shape conjunctions from Section 4.2, showing that they exhibit a compositional structure – a property that, as we will show, has direct consequences for model behavior. We then turn to a denser analysis of color representations, extracting centroid-based vectors for a fine-grained set of hues and characterizing their continuous geometry. Additional figures spanning all three models are provided in Appendix E.

## 5.1. Color-Shape Compositionality

The success of color steering across objects of varied shapes in Section 4.1 already suggests that color and shape are encoded independently: a single color vector that reliably manipulates perception regardless of the object's shape must be capturing an uncorrelated direction. We now verify this directly by examining the geometric relationships between all 36 color-shape concept vectors. Figure 3 shows their pairwise cosine similarities for Gemma.

The structure is strikingly compositional: objects sharing a color or shape are more similar than those sharing neither, and – crucially – similarity depends on *whether* two objects share a feature, not on *which* feature they share. This is visible in the cleanly separated distributions at the bottom of Figure 3: "same color" and "same shape" pairs form distinct, non-overlapping clusters, both well-separated from "neither" pairs. This clean factorization into independent color and shape dimensions holds across all three models and all extraction methods (see Appendix E), suggesting it reflects a fundamental organizational principle rather than an artifact of our methodology.

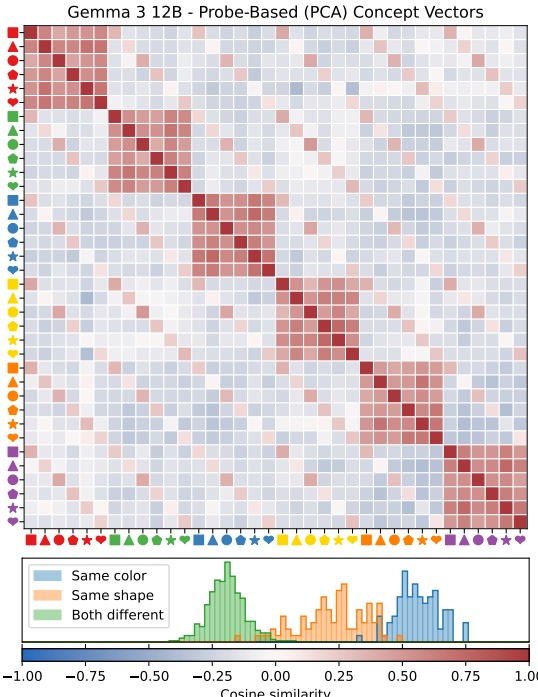

*Figure 3.* Matrix of cosine similarities between 36 color-shape concept vectors (Gemma). The block structure reflects shared colors (large blocks) and shapes (sub-diagonals). Bottom: similarity distributions for object pairs sharing color, shape, or neither.

## 5.2. Semantic Structure in Color Representations

The compositional analysis above treated colors as discrete symbols. However, to understand the fine-grained mechanics of interference, we must analyze the continuous geometry of the color space. To this end, we generate concept vectors for 100 hues sampled uniformly around the HSV color wheel using the centroid-based extraction method.

**Continuous color space.** Figure 4(a-b) reveals that hue representations form a continuous 1-dimensional manifold (Engels et al., 2025) that is circular in topology but geometrically non-trivial. Unlike a perfect HSV circle, the manifold exhibits *semantic warping*: hues linguistically categorized as "green" cluster tightly together, distorting the metric space to align with natural language categories. This confirms that the VLM's visual representation is not a faithful physical map, but a *semantic* map warped by the discrete token space of the LLM.

This combination of within-category compression and between-category separation is the geometric signature of the "accordion effect" associated with categorical perception (Harnad, 1990); to our knowledge, such structure has not previously been characterized in the latent space of VLMs.

**Semantic Similarity profiles.** Given a hue value $h$ and its corresponding normalized concept vector $\hat{\mathbf{v}}(h)$, we define

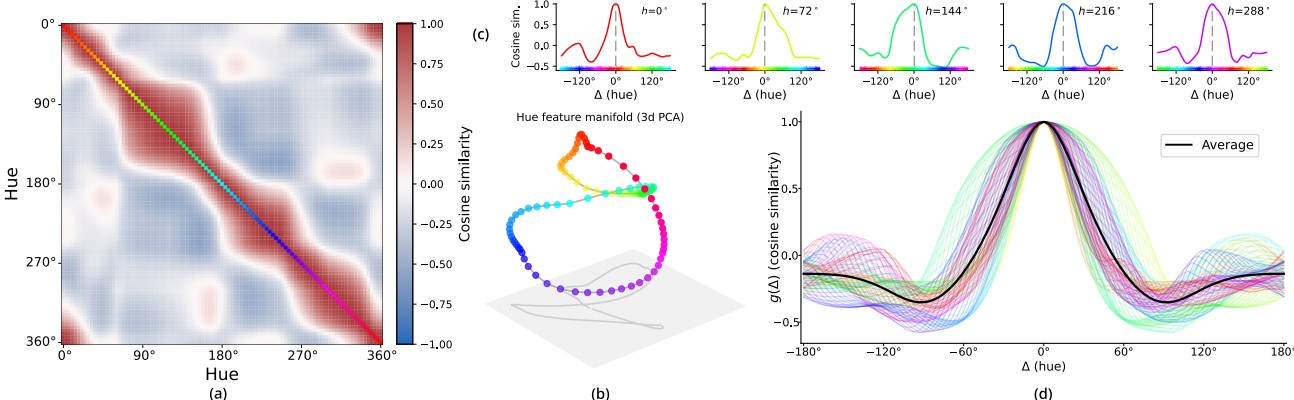

*Figure 4.* (a) Heatmap of the cosine similarities between centroid-based hue concept vectors found in the vision embeddings of Qwen. (b) Projection of color representations in the first 3 principal components. (c,d) Semantic Similarity Function $g_h(\Delta)$ for different hues ($h$ corresponds to the color of the curve). The black line represents the average function $g(\Delta)$.

the *Semantic Similarity Function* as:

$$g_h(\Delta) = \hat{\mathbf{v}}(h)^\top \hat{\mathbf{v}}(h + \Delta) \tag{4}$$

which measures the cosine similarity between representation directions as a function of semantic displacement $\Delta$. These semantic similarity profiles allow us to analyze the *resolution* of the color representation manifold (Figure 4 c-d). We find that these profiles mirror generalization gradients in cognitive science (Shepard, 1958), identifying a distinctive interaction profile that explains the tension between generalization and discrimination:

1. **Local Generalization** ($|\Delta| < 90°$)**:** We observe a monotonic decay in similarity, consistent with observations by Modell et al. (2025) who hypothesize that such local structure is necessary to capture an adequate notion of distance. This continuity allows the model to generalize across shades (e.g., binding "crimson" and "scarlet" to the same concept).

2. **Distal Interference** ($|\Delta| > 90°$)**:** Unexpectedly, similarity decays to a minimum but rises again in distant regions (the "ripples"). This "Mexican Hat" profile (Müller et al., 2005) is reminiscent of the "rippled representations" observed in LLM counting features (Gurnee et al., 2026). These non-monotonic "noisy" patterns of similarity in the tails of the similarity function suggest the presence of a finite representational resolution as discussed in Nurisso et al. (2026).

This profile provides a mechanistic basis for the "Curse of Generalization" (Frankland et al., 2026): the same geometric property that enables continuity between similar shades precludes orthogonality between distinct ones. This creates **geometric interference** which lays the groundwork for the binding failures observed in multi-object scenes.

**Universality of Geometric Constraints.** Is this interference pattern an artifact of a specific model? To test this,

we performed a Representational Similarity Analysis (RSA) (Kriegeskorte et al., 2008) across Qwen, InternVL, and Gemma. As shown in Table 3, the geometries are nearly identical (all pairwise correlations $r > 0.93$). This suggests that the geometry profiles and the interference they cause are a universal constraint arising from mapping continuous visual signals into discrete semantic spaces.

|  | QWEN | INTERNVL | GEMMA |
|---|---|---|---|
| QWEN | 1. | | |
| INTERNVL | 0.93 | 1. | |
| GEMMA | 0.94 | 0.98 | 1. |

*Table 3.* Representational similarity analysis (RSA) across the three models. Each entry shows the Pearson correlation between the models' representational similarity matrices, indicating strong alignment of representational geometry.

## 6. Geometric Interference and VLM Behavior

We now show that the geometric organization characterized in Section 5 correlates with model behavior in two tasks: a visual search paradigm widely studied in cognitive psychology, and a color similarity evaluation task. In both cases, cosine similarity between concept vectors (which in turn were obtained from single-object images) accounts for systematic patterns of model errors on complex multi-object scenes.

### 6.1. VLM Errors in Visual Search Task

The compositional structure documented above makes a clear prediction: if concept vectors for objects sharing a feature (e.g., same color or same shape) are geometrically closer, then visual search should be harder when distractors share features with the target. We test this prediction using a visual search paradigm adapted from Campbell et al. (2024).

**Task and stimuli.** We generate synthetic images containing 2 to 40 colored shapes and query whether a specific color-shape conjunction is present (Figure 5). We call the queried item the *target* and the other items *distractors*. For each image, we compute the maximum cosine similarity between the target's centroid-based concept vector and those of all distractor objects – a measure of how "confusable" the hardest distractor is. We then bin trials by this similarity and compute accuracy within each bin.

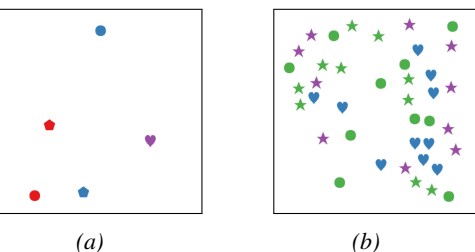

*(a)*          *(b)*

*Figure 5.* Visual search task. The query reads "`Is there a purple heart in the image? Answer YES or NO`". (a) Target present with dissimilar distractors. (b) Target absent but distractors share features with target (purple star, blue heart), creating high interference.

**Results.** Figure 6 shows Qwen's accuracy as a function of the maximum similarity between the centroid-based concept vectors associated with the target and the distractor shapes; results for all models are summarized in Table 4. Across all three VLMs, accuracy decreases monotonically with distractor similarity, yielding strong negative correlations for both target-present trials ($r = -0.90$ to $-0.97$) and target-absent trials ($r = -0.69$ to $-0.88$). The effect is remarkably consistent: the geometry of the representation space – measured entirely from synthetic single-object images – predicts error rates on a complex multi-object task.

Geometric interference manifests as opposing error types across conditions. When the target is present, high-similarity distractors cause the model to miss it (false negatives); when the target is absent, they cause hallucinations (false positives, or "illusory conjunctions"). Both failure

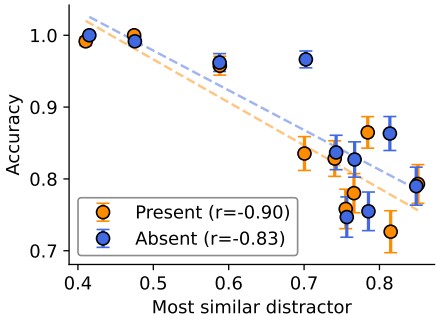

*Figure 6.* Visual search accuracy for Qwen decreases with distractor similarity. Each point represents accuracy for trials binned by maximum cosine similarity between the target and distractor concept vectors. Orange: target present. Blue: target absent.

modes trace to the same geometric cause: interference between overlapping concept vectors in a shared representational space.

| | QWEN | INTERNVL | GEMMA |
|---|---|---|---|
| TARGET PRESENT | $-0.90$ | $-0.90$ | $-0.97$ |
| TARGET ABSENT | $-0.83$ | $-0.88$ | $-0.69$ |

*Table 4.* Pearson correlations between maximum distractor similarity and model accuracy on visual search. Strong negative correlations indicate that geometric interference predicts task difficulty.

### 6.2. VLM Errors in Similarity Task

The semantic structure documented in Section 5.2 – where representational similarity deviates systematically from perceptual distance – makes a testable prediction: model confidence should track *representational* similarity between colors, not their distance in color space. We test this using the color similarity task from Nurisso et al. (2026).

**Task.** Models are shown two images (Figure 7): a setup image containing 4–12 colored squares labeled with letters, and a query image with a single target color. The model must identify which labeled color is most similar to the target. This task is harder when multiple colors in the setup image are close to the query color, creating interference.

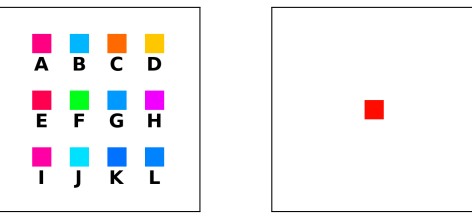

*Figure 7.* Color similarity task. Left: setup image with labeled colors. Right: query color.

**Quantifying model confidence.** To measure the model's internal confidence, we examine the output logits for answer tokens ("A", "B", etc.), select the top-scoring token $I = \underset{i=1,\dots,N}{\arg\max}\, l_i$ and compute the *logit separation*:

$$\text{logit sep} = l_I - \frac{1}{N-1}\sum_{i \neq I} l_i \qquad (5)$$

High logit separation indicates confident decisions; low separation indicates uncertainty.

We compare two predictors of this confidence. Given the query color $c_Q$ and setup colors $c_1, \dots, c_N$, we define *similarity separation* as:

$$\text{sim-sep} = g(c_I, c_Q) - \frac{1}{N-1}\sum_{i \neq I} g(c_i, c_Q) \qquad (6)$$

where $I$ is the model's chosen answer and $g$ is a similarity function. When similarity separation is high, one color

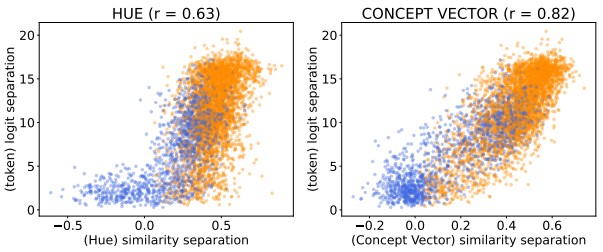

*Figure 8.* Similarity separation vs. logit separation for InternVL. Left: HSV-based similarity. Right: concept vector similarity. Each point is one trial; color indicates whether the model's answer matches the closest item under that similarity metric (orange/yes, blue/no). Concept vectors yield tighter correlation, indicating they better capture the model's internal similarity structure.

stands out as clearly most similar; when low, multiple colors compete. We compare two choices for $g$: (i) linear decay in HSV distance, and (ii) cosine similarity between concept vectors.

**Results.** Figure 8 shows the relationship between similarity separation and logit separation for InternVL; full results appear in Table 5. The key finding is that concept vector similarity predicts model confidence far better than perceptual color distance. Across all three models, concept vectors achieve correlations of $r = 0.78$-$0.84$ with logit separation, compared to $r = 0.60$-$0.66$ for HSV-based similarity.

Notably, both metrics predict the model's *answer* with similar accuracy (70–75%): concept vectors score slightly higher for Qwen and Gemma, while HSV hue wins marginally on InternVL. Yet concept vectors far better predict *confidence*. This dissociation is revealing: the model's decisions roughly track perceptual similarity, but its uncertainty is governed by representational geometry. The block structure visible in Figure 4 – where linguistically similar colors cluster regardless of HSV distance – directly shapes how the model experiences ambiguity.

## 7. Conclusions

This paper offers a mechanistic insight for VLM failures in multi-object visual tasks: interference between concept vectors in a shared latent space. Our findings support the "Curse of Generalization" hypothesis (Frankland et al., 2026): representational structures required for flexible generalizable reasoning induce vulnerability to interference. Thus, binding failures are the cost of compressing rich visual information into reusable representations (e.g., Conklin et al. 2026; Assouel et al. 2025).

The consistency of our findings across architecturally diverse models points to fundamental constraints of the current VLM paradigm rather than artifacts of any specific architecture: all three exhibit compositional representations with similar geometric structure, error rates that scale with

|  | QWEN | INTERNVL | GEMMA |
|---|---|---|---|
| HUE CORRELATION | 0.664 | 0.627 | 0.597 |
| CB-CV CORRELATION | 0.843 | 0.822 | 0.777 |
| HUE/VLM ACCURACY | 72.3% | 71.6% | 67.1% |
| CB-CV/VLM ACCURACY | 74.6% | 70.9% | 69.6% |

*Table 5.* Predicting VLM behavior on the similarity task. Top rows: Pearson correlation between similarity separation and logit separation. Bottom rows: accuracy of predicting which color the VLM selects. Concept vectors (CB-CV) predict confidence substantially better than HSV hue distance, despite similar accuracy in predicting the chosen answer.

interference, and semantic organization aligned with linguistic categories. We demonstrated that concept vectors form structured manifolds the geometries of which are not merely descriptive but predictive: cosine similarity between vectors correlates strongly with model confidence ($r > 0.77$) and accuracy ($|r| > 0.83$) on downstream tasks. Finally, our steering experiments confirm that these vectors are causally active: directions derived from synthetic data reliably manipulate perception in natural images (85-96% accuracy), demonstrating that the geometry we measure directly shapes behavior.

The parallel to human cognition is notable: Treisman's feature integration theory (Treisman & Gelade, 1980) proposed that biological vision solves binding through serial attention. Our findings suggest that a similar problem is faced by artificial systems, and for a similar reason: the system is required to keep potentially confusable representations distinct from one another, and a failure to do so gives rise to "illusory conjunctions." This reflects the curse of generalization: interference arises from shared, generalizable representations. Thus, systems that favor generalization face a tradeoff between generalization and processing capacity – that is, the ability to simultaneously process multiple distinct representations. A related parallel to human cognition concerns the color manifold itself, for which the within-category compression and between-category separations echo the "accordion effect" of human categorical perception (Harnad, 1990; Goldstone, 1994). To our knowledge, this is among the first such observations in a VLM's latent space. We make no claim to demonstrating categorical perception as a perceptual phenomenon; whether this structure is induced by the LLM's discrete token space or arises intrinsically remains open for future work.

**Limitations.** Our framework relies on synthetic datasets tractable for simple concepts but difficult to scale to abstract attributes. We analyzed only the vision output; language representations may account for unexplained failure modes. We characterized *what* representations look like but not *how* they are computed. A complete account requires tracing the underlying circuits.

## Acknowledgements

The work reported in this article was supported in part by funding from a Vannevar Faculty Fellowship to JDC, and an NSF GRFP and Princeton AI Lab Natural and Artificial Minds Graduate Fellowship to IDC.

## Impact Statement

This paper presents work whose goal is to advance the understanding of Vision-Language Model failures through mechanistic interpretability. We believe this research has predominantly positive societal implications:

**Benefits for AI Safety and Reliability.** By identifying the geometric signatures that predict VLM errors – such as hallucinations and binding failures – this work provides a principled framework for anticipating when models are likely to fail. This could inform the development of more robust systems and enable better calibration of model confidence in safety-critical applications (e.g., medical imaging, autonomous systems).

**Transparency and Interpretability.** Our methods for extracting and validating concept vectors contribute to the broader goal of making AI systems more interpretable. Understanding why models fail, rather than merely documenting that they do, is essential for responsible deployment.

**Potential Concerns.** The steering techniques we demonstrate could, in principle, be used to adversarially manipulate model outputs. However, our interventions require white-box access to model activations and are primarily diagnostic in nature. We believe the interpretability benefits substantially outweigh this limited risk.

**Cognitive Science Implications.** The parallels we draw between VLM failures and human binding errors may inform theories of biological vision, though care should be taken not to over-anthropomorphize artificial systems.

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

## A. Synthetic Datasets Construction Details

All synthetic datasets are generated by sampling objects from combinations of 6 colors (red, green, blue, yellow, orange, purple) and 6 shapes (square, triangle, circle, pentagon, star, heart), rendered on a white background with non-overlapping objects (see examples in Figures 1 and 5). All synthetic images have resolution $448 \times 448$.

**Visual search dataset.** This kind of dataset is used both for training the probes described in Sections 4.1 and 4.2, as well as in the visual search experiment described in Section 6.1. For each image, we first select a target object $T = (c_t, s_t)$. We then populate the scene with $N_{dist}$ distractor objects, where $N_{dist} \in \{2, 4, 8, 16, 24, 32, 40\}$. The target object is represented only in half of the images.

We control the degree of feature overlap between distractors and the (possibly absent) target via a parameter $P_{\text{int}} \in \{0, 0.25, 0.5, 0.75, 1\}$, defined as the fraction of distractors sharing exactly one property (color or shape) with the target. This constraint is enforced even in target-absent images; it ensures that when $P_{\text{int}} = 1$, every distractor is a "near-miss" (e.g., sharing color or shape), creating maximal geometric crowding. Conversely, when $P_{\text{int}} = 0$, all distractors are feature-disjoint from the target, minimizing interference.

Distractors are sampled such that (i) each image contains 4 unique color-shape combinations (except when $N_{\text{dis}} = 2$), and (ii) the proportion of distractors sharing the target's color and those sharing its shape is kept as balanced as possible. We generate 4 images for each combination (target object, $N_{\text{dis}}$, $P_{\text{int}}$) (2 with and 2 without the target), yielding a total of 4752 images.

**Centroid-based concept vector dataset.** To estimate concept centroids for color-shape combinations, we generate images containing a single colored shape placed at controlled spatial locations. We define 9 canonical positions (center, 4 corners, and 4 cardinal directions) and apply small perturbations to each, yielding 81 distinct positions. For each of the 36 color-shape combinations, we generate one image per position, for a total of 2916 images.

**Color extraction dataset.** To study the geometry of color representations, we generate images containing a single centered square (Figure 1b). We sample 100 distinct colors uniformly over hue, producing a dataset that isolates color from shape and spatial variability.

## B. Experimental Details: Color Steering in Natural Images

This appendix describes in more detail how the experimental results reported in Section 4.1 were obtained. The experiment consists in comparing representations for 6 colors obtained by the three methods described in Section 3.2. Here we will clarify how probes were trained, their achieved accuracy on the training task, and the steering procedure adopted. Finally, we describe a preliminary experiment that applies the same concept to shape steering.

### B.1. Probe Training and Accuracy

**Dataset** We train 6 probes, each predicting the presence of a specific color in the visual search dataset from the activations collected at the output of the multimodal projection layer. An independently generated but analogous dataset is used as test set; as shown in Tables 6 and 7, probes achieve near-optimal accuracy on both training and test set.

|           | QWEN | INTERNVL | GEMMA |
|-----------|------|----------|-------|
| TRAIN SET | 100% | 99.8%    | 99.7% |
| TEST SET  | 100% | 99.8%    | 99.6% |

*Table 6.* Color probe accuracies (aggregated on all 6 colors).

**Optimization** All probes are trained using stochastic gradient descent with a learning rate of $10^{-3}$ and batch size 50. We employ a learning rate scheduler (`ReduceLROnPlateau` in PyTorch) to adapt the learning rate based on the training loss. Training is stopped early based on convergence of the training loss.

| | | | | | | | | | | | | |
|---|---|---|---|---|---|---|---|---|---|---|---|---|
| | *(a)* Train Set | | | | | | *(b)* Test Set | | | | | |
| | RED | GREEN | BLUE | YELLOW | ORANGE | PURPLE | RED | GREEN | BLUE | YELLOW | ORANGE | PURPLE |
| QWEN | 100.0% | 100.0% | 100.0% | 100.0% | 100.0% | 100.0% | 100.0% | 100.0% | 100.0% | 100.0% | 100.0% | 100.0% |
| INTERNVL | 99.8% | 100.0% | 99.9% | 99.6% | 99.6% | 99.8% | 99.9% | 99.9% | 99.9% | 99.8% | 99.6% | 99.9% |
| GEMMA | 99.3% | 99.9% | 99.7% | 99.7% | 99.6% | 99.7% | 99.3% | 99.9% | 99.7% | 99.8% | 99.3% | 99.9% |

*Table 7.* Color probes accuracies.

## B.2. Natural Image Steering Procedure

Each image in the steering dataset depicts a single foreground object associated with a predominant color chosen from the set {red, green, blue, yellow, orange, purple}.

We first query the model to identify the object's color using the following prompt:

```
Observe the image carefully. What color is the {object_name}?
Answer in one word.
```

We retain only those instances for which the model's response matches the ground-truth color. For these filtered samples, we then reissue the prompt while applying our steering procedure, removing the true color representation and injecting an alternative target color.

In evaluating the responses, we accept any capitalization variant of the six color names as correct, while rejecting more specific shade descriptions (e.g., "lime").

## B.3. Additional Experiment: Shape Steering

While the main text focuses on color steering, shape is a natural candidate for an analogous experiment. However, shape presents fundamental challenges that motivated our focus on color: the shape of natural objects is often ill-defined or context-dependent, and constructing a balanced dataset is considerably harder. We nonetheless include a preliminary shape steering experiment as a point of comparison.

We use natural images from the VQA/COCO dataset (Goyal et al., 2017), filtering for questions of the form "What shape is ..." whose answer matches one of our six probed shapes, yielding 297 examples (square: 126, circle: 75, triangle: 47, heart: 28, star: 20, pentagon: 1). As in the color steering experiment, we steer only examples the model answers correctly (183, 177, and 111 examples for Qwen, InternVL, and Gemma, respectively), applying steering once per alternative shape.

| | QWEN | INTERNVL | GEMMA |
|---|---|---|---|
| PROBE | 0.3% | 2.6% | 0.2% |
| PCA PROBE | 9.3% | 22% | 0.9% |
| CENTROID | 19.7% | 1.1% | 25.8% |

*Table 8.* Steering of Shapes on Natural Images – Success Rate

As shown in Table 8, results are weaker than color steering, yet remain non-trivial: Qwen and Gemma reach 19.7% and 25.8% with centroids, while InternVL reaches 22.0% with PCA probes – consistent with the pattern observed in Tables 1 and 2.

The weaker performance may reflect a fundamental asymmetry between color and shape as visual attributes. Color is a physical property intrinsic to an object's surface, largely context-independent, and continuously grounded in perception. This is why concept vectors extracted from synthetic stimuli transfer robustly to natural images: synthetic and natural color distributions share the same underlying perceptual structure. Shape, by contrast, is inherently categorical and context-sensitive. There is a semantic gap between *being* a square (an object identity) and *having* a square shape (a geometric property): a pool can be "round," a road sign "triangular," a cloud "heart-shaped." Synthetic shapes – uniform, isolated, and unambiguous – are too far removed from the variability of natural images, and the extracted directions do not capture the richer, more distributed representations that natural shape concepts require.

Consistently with this interpretation, shape steering accuracy depends strongly on both the source and target shape identity, unlike color steering (see Table 9).

| SHAPE | QWEN | INTERNVL | GEMMA |
|---|---|---|---|
| SQUARE | 1% | 0% | 2% |
| TRIANGLE | 3% | 0% | 42% |
| CIRCLE | 1% | 1% | 5% |
| PENTAGON | – | – | 40% |
| STAR | 86% | 4% | 65% |
| HEART | 68% | 3% | 41% |

*(a)* Original Shape

| SHAPE | QWEN | INTERNVL | GEMMA |
|---|---|---|---|
| SQUARE | 36% | 1% | 35% |
| TRIANGLE | 28% | 3% | 27% |
| CIRCLE | 25% | 1% | 39% |
| PENTAGON | 11% | 0% | 13% |
| STAR | 14% | 1% | 15% |
| HEART | 13% | 1% | 31% |

*(b)* End Shape

*Table 9.* Shape steering accuracy for centroid-based concept vectors by original shape (a) and end shape (b).

## C. Experimental Details: Steering Compositional Concepts

This appendix provides full experimental details for Section 4.2, including probe training, the steering evaluation protocol, and additional experiments extending the results of the main text.

### C.1. Probe Training and Accuracy

We train a separate probe for each color-shape combination (36 in total), following a supervised discrimination approach as described in Section 3.2. Our dataset construction leverages the multi-object structure of the visual search dataset to expose probes to both single and multiple occurrences of each concept, while systematically varying the presence of objects that share features with the probe target.

**Training set construction.** Given a target concept (e.g., "red square"), we construct the training set by aggregating three types of images from the visual search dataset:

- **Target-present images:** all images where the selected concept is the designated target. These already provide balanced positive and negative labels (target present vs. absent).

- **Distractor images:** all images where the concept appears as a distractor. These are labeled as positive examples.

- **Negative samples:** a set of images, randomly sampled to match the number of distractor images, where the concept does not appear and is not the target. These are labeled as negative examples.

This construction ensures that the probe is trained to detect the presence of a concept regardless of whether it is task-relevant (target) or not (distractor), while maintaining balance between positive and negative labels. It yields a set of approximately 1100 images for each probed objects.

**Optimization.** All probes are trained using stochastic gradient descent with a learning rate of $10^{-3}$. We additionally employ a learning rate scheduler (`ReduceLROnPlateau` in PyTorch) to adapt the learning rate based on the training loss. Training is stopped early based on convergence of the training loss.

**Test set construction.** We construct the probe evaluation set by generating a new visual search dataset following exactly the same procedure described in Appendix A, while using independent random sampling. This ensures that the test distribution matches the training distribution while avoiding any overlap at the image level.

| | QWEN | INTERNVL | GEMMA |
|---|---|---|---|
| TRAIN SET | 99.9% | 98.0% | 99.0% |
| TEST SET | 99.9% | 98.0% | 98.7% |

*Table 10.* Train and test set accuracy for Attention Pooling probes (all probing targets aggregated, baseline 50%).

## C.2. Steering Procedure

We define the following intervention procedure, which makes use of a new set of synthetic images generated using geometric shapes as "objects":

1. generate an image containing at least two kinds of object, $A$ and $C$, but not object $B$; make sure that no property is shared among these three objects (three different colors and three different shapes);

2. verify that the model, without applying any steering operation on its internal activations, correctly recognizes the presence of objects $A$ and $C$ and the absence of object $B$;

3. generate a new image, identical to the previous one but where object $A$ has been substituted with $B$; verify that in this new image the model correctly finds $B$ and $C$ and recognizes that $A$ is absent;

4. go back to the first image, containing $A$ but not $B$, and repeat the prompting, this time applying the steering procedure (3); if the steering is successful, we expect the model to give the same answers obtained at point 2.

If any of the points 2 or 3 fails, we try to generate a new image, since at this stage we are only interested in the effectiveness of our steering procedure; after 10 unsuccessful attempts, we exclude the triple $(A, B, C)$ from the experiment.

To select the triple $(A, B, C)$ and the remaining objects in the image, we proceed as follows. For each combination of target shape $A$ (36 candidate shapes), number of distractors $N_{\text{dis}} \in \{4, 16, 40\}$, and interference probability $P_{\text{int}} \in \{0.25, 0.5, 0.75\}$, we sample 2 pairs of shapes $(B, C)$ with $A, B, C$ all distinct. Each pair yields two images: one containing shape $A$ (but not $B$) among $N_{\text{dis}}$ distractors, and one where $A$ and $B$ are swapped. Each image is filled with additional shapes so that the total number of objects reaches $N_{\text{dis}}$, with a proportion $P_{\text{int}}$ of distractors sharing the shape or color of object $A$. This gives $36 \times 3 \times 3 \times 2 \times 2 = 1{,}296$ images in total.

The prompt explicitly queries the presence of a specific color-shape combination, constraining the model's output to a binary (yes/no) response:

```
Look at the image carefully.
Is there a {color} {shape} in the image?
Answer only 'yes' or 'no'.
```

Model answers are generated greedily by selecting the highest-scoring token. We accept any capitalization of *yes* and *no* as valid responses; no other outputs were observed.

## C.3. Additional Experiment: Mean Aggregation Probes

To evaluate whether a different probing architecture could provide better representations, we repeat the experiments of Section 4.2 with concept vectors extracted from mean aggregation probes. Using the notation of Section 3, the probe is defined as:

$$\overline{\mathbf{h}} = \frac{1}{L} \sum_{t=1}^{L} \mathbf{h}_t$$
$$\hat{y} = \sigma \left( \overline{\mathbf{h}}^\top \mathbf{u}_c + b_{out} \right) \tag{7}$$

We train the probe with the same procedure described in Appendix C.1; as shown in Table 11, these probes show a lower accuracy on the training task than those using attentive pooling.

After training, we obtain the concept vector for the color-shape combination $c$ as the normalized version of parameter $\mathbf{u}_c$: $\hat{\mathbf{v}}_{mean}^{(c)} = \mathbf{u}_c / \|\mathbf{u}_c\|$.

Table 12 shows steering accuracies for all extraction methods, including those already reported in Table 2 for comparison. Mean aggregation probes consistently outperform unregularized attention pooling probes; however, they do not achieve the highest steering accuracy for any of the three models. PCA regularization appears less effective for mean aggregation probes, and even degrades steering accuracy on Qwen.

|  | QWEN | INTERNVL | GEMMA |
|---|---|---|---|
| TRAIN SET | 88.8% | 73.8% | 89.5% |
| TEST SET | 88.3% | 73.2% | 88.8% |

*Table 11.* Train and test set accuracy for Mean Aggregation Probes (all probing targets aggregated, baseline 50%).

|  | QWEN | INTERNVL | GEMMA |
|---|---|---|---|
| PROBE | 0.0% | 2.0% | 0.0% |
| PCA PROBE | 44.2% | **46.8%** | 17.9% |
| CENTROID | **78.1%** | 35.9% | **75.3%** |
| MEAN AGG PROBE | 53.5% | 2.9% | 10.4% |
| PCA MEAN AGG PROBE | 37.8% | 3.0% | 20.8% |

*Table 12.* Synthetic image steering results across models and probe types. Probe refers to the attention pooling probing architecture described in Section 3.2.

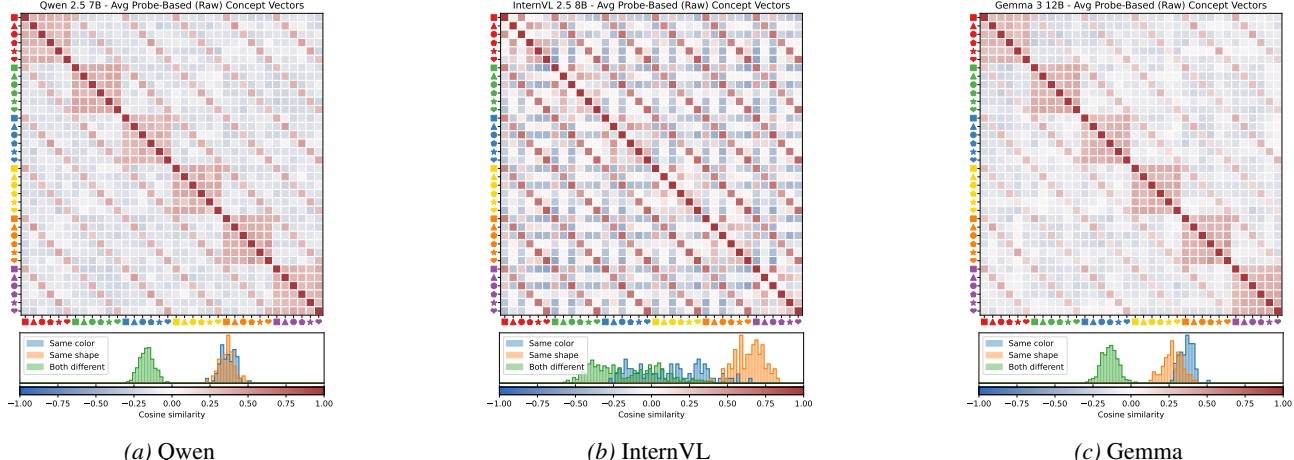

*(a) Qwen*      *(b) InternVL*      *(c) Gemma*

*Figure 9.* Matrices of cosine similarities between concept vectors extracted through Mean Aggregation Probes and distribution of values for three VLMs.

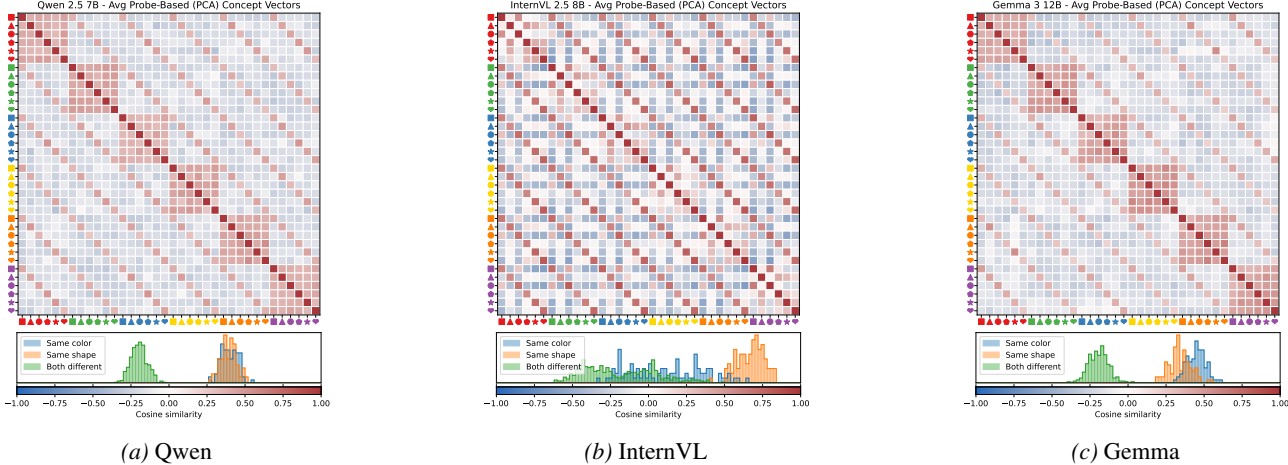

*(a) Qwen*      *(b) InternVL*      *(c) Gemma*

*Figure 10.* Matrices of cosine similarities between concept vectors extracted through Mean Aggregation Probes with PCA-based regularization and distribution of values for three VLMs.

# D. Visual Search Task Details

To evaluate the impact of representational geometry on multi-object reasoning, we developed a synthetic visual search task adapted from Campbell et al. (2024). The images employed in the experiment come from the visual search dataset; by construction, for each image a target object $T$ is defined.

## D.1. Task Protocol

The dataset is balanced such that the target $T$ is present in 50% of trials (added to the $N_{\text{dist}}$ distractors) and absent in the remaining 50%. We queried the VLM using the binary classification prompt:

```
Look at the image carefully. Is there a [color] [shape] in the image?
Answer only 'yes' or 'no'.
```

Performance was evaluated via exact string matching (case-insensitive).

## D.2. Quantifying Geometric Interference

To validate the impact of this generation protocol, we compute a *Distractor Similarity Score* for every generated image. Let $\mathbf{v}_T$ be the centroid-based concept vector for the target. The interference level $I$ is defined as the maximum cosine similarity between the target and any distractor $d$ in the scene:

$$I = \max_{d \in \mathcal{D}} \cos(\mathbf{v}_T, \mathbf{v}_d) \tag{8}$$

We confirm in Section 4.4 that trials with high $p_{int}$ yield high geometric interference $I$, which strongly correlates with model error rates.

# E. Extended figures

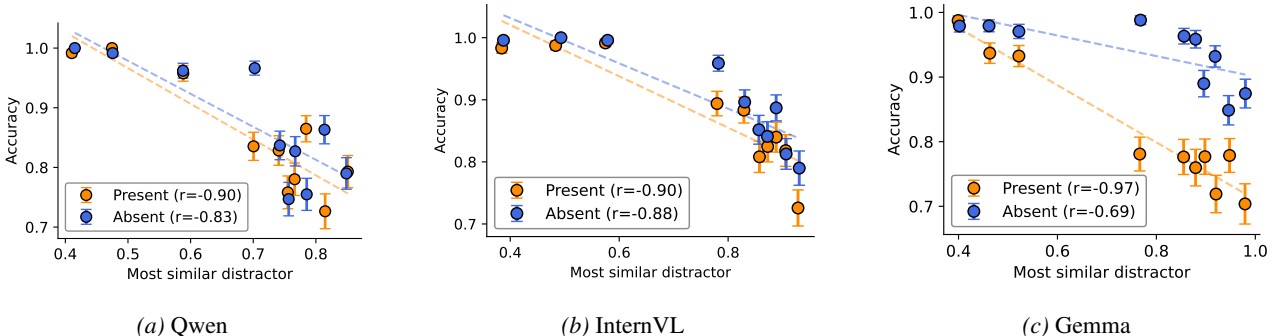

*(a)* Qwen  *(b)* InternVL  *(c)* Gemma

*Figure 11.* Extension of Figure 6: visual search accuracy against distractor similarity (computed through centroid-based concept vectors) for all three models.

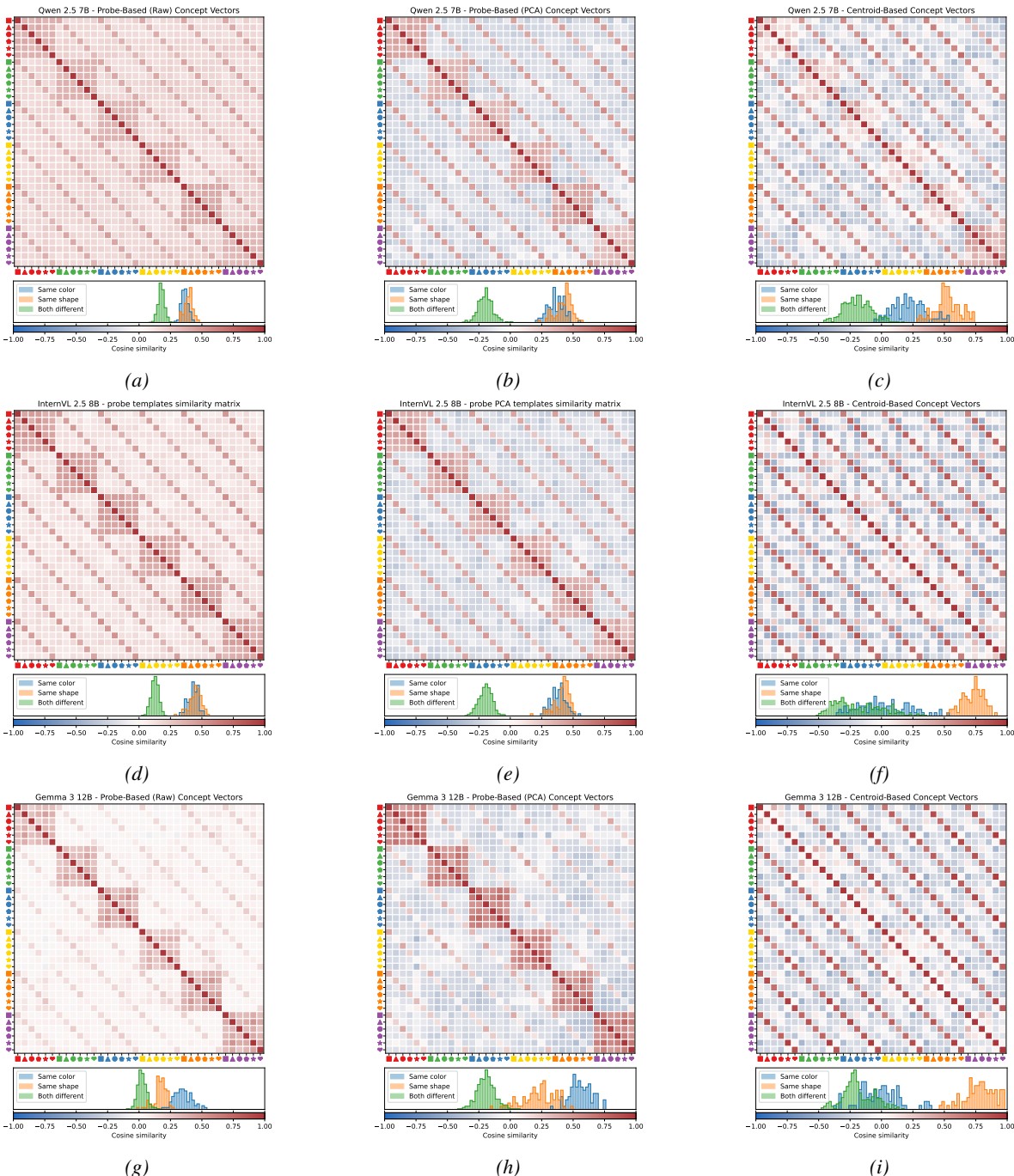

*Figure 12.* Matrix of cosine similarity between concept vectors related to colored shapes, extracted from vision embeddings of different models for all three template extraction methods. Below the similarity matrix, we plot the distribution of similarity values for object pairs, grouped by whether the objects share color, share shape, or share neither property.

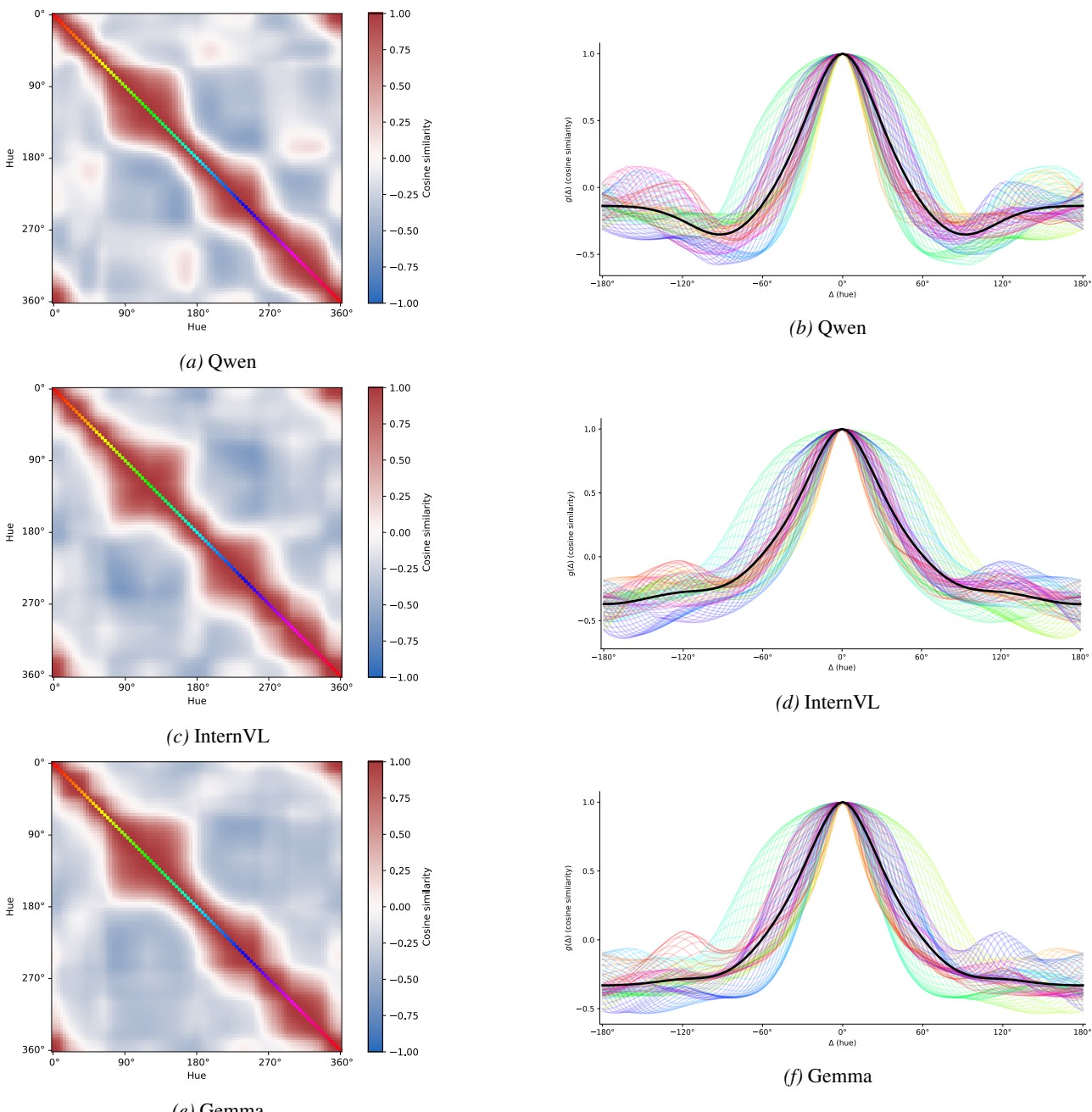

*Figure 13.* Matrices of cosine similarities between contrastive templates representing color hues, in three VLM models, and their corresponding average similarity function.

