# OpenReview forum: "The Geometry of Representational Failures in Vision Language Models"
_ICML.cc/2026/Conference — ICML 2026 regular_

### Official Review · Reviewer_Dx4K · 2026-02-17

**Soundness:** 4
**Presentation:** 3
**Significance:** 4
**Originality:** 4
**Overall Recommendation:** 6
**Confidence:** 3

**Summary:**

This paper studies how the binding of simple visual features is encoded in 3 vision-language models (VLMs) through a set of quantitative and qualitative experiments using steering and geometric methods applied to concept vectors extracted from VLM image token embeddings. The results suggest that the models encode combinations of features compositionally, that the representational geometry of concept vectors form a manifold whose structure is predictive of model performance on downstream tasks, and that compositional representations also lead to semantic interference effects that predict attested model hallucination issues.

**Compliance With Llm Reviewing Policy:**

Affirmed.

**Final Justification:**

This was a very strong paper to start with, dealing with central topics in model interpretability and cognitive science, and my opinion remains very positive after the rebuttal.

**Key Questions For Authors:**

1) I had problems understanding exactly which representations where used to construct the concept vectors: were these simply the outputs of the module mapping vision to LLM tokens?

2) You might want to draw a connection between your observation of how language affects the perceptual manifold and "categorical perception" (https://en.wikipedia.org/wiki/Categorical_perception) in cognitive science.

**Limitations:**

yes

**Strengths And Weaknesses:**

Strengths:

The paper deals with central themes in AI and cognitive science, and it presents a large number of original and convincing experiments.

Weaknesses:

I honestly don't see any serious weakness. I can only say that the paper was a bit hard to follow, due to its slightly unusual narrative structure, and for the sheer number of experiments that are presented, that are only superficially described in the main text. However, I'd say this is a natural price to pay for such a rich article.

---

> ### Author Rebuttal · Authors · 2026-03-31
>
> We thank the reviewer for pointing us to categorical perception (CP) - we had not previously connected our findings to the categorical perception framework explicitly, and we agree it is highly relevant to our findings.
>
> Our observation that the hue manifold is "semantically warped" - with linguistically co-categorized hues clustering together regardless of their HSV distance - is precisely the signature of CP: within-category compression and between-category expansion of perceptual distances. We had approached this phenomenon through related concepts that are already present in the paper: Shepard's (1958) generalization gradients, Berlin \& Kay's (1969) universality of color categories, and Zaslavsky et al.'s (2018) efficient compression framework. These references collectively describe the same underlying mechanism that CP formalizes, so the conceptual bridge was implicit rather than explicit in our current draft.
>
> We will add an explicit discussion of CP in the revised manuscript. This addition will strengthen the paper in two ways. First, it connects our geometric finding - that concept vector similarity predicts model confidence better than physical HSV distance - to a well-established empirical literature in psychophysics and cognitive science. Second, it sharpens the claim that VLMs have emergently acquired a human-like perceptual organization, rather than a truthful map of the physical stimulus space. We will also note the parallel to the linguistic relativity topic discussed in the CP literature, which aligns with our interpretation that the LLM's discrete token space distorts the visual manifold.
>
> The reviewer also asks to clarify "which representations were used to construct the concept vectors". These are exactly the output of the model mapping vision to LLM tokens. The complete pipeline is as follows: the vision encoder maps the input image to a sequence of visual token embeddings, which are then projected into the LLM's embedding space via the learned alignment layer. These projected tokens are injected into the LLM and processed within its residual stream alongside text token embeddings. Since all cross-modal interaction and task-relevant computation occur inside the LLM, we focus our analysis there - specifically on the activations at the positions corresponding to visual tokens, within the LLM's residual stream. The dimensionalities of the concept vectors (d = 3584 for Qwen, 4096 for InternVL, 3840 for Gemma) reflect the LLM hidden dimensions, confirming that these are not the raw ViT output features.
>
> We will make this information more explicit in the final manuscript.

---

> > ### Author Rebuttal · Reviewer_Dx4K · 2026-03-31
> >
> > Thanks for clarifying my doubts.

---

> > > ### Author Response · Authors · 2026-04-08
> > >
> > > We sincerely thank the reviewer for the thorough engagement with our rebuttal.
> > > We are glad our responses addressed the main concerns, and we will incorporate all the discussed improvements in the revision.

---

### Official Review · Reviewer_b3fs · 2026-03-11

**Soundness:** 3
**Presentation:** 2
**Significance:** 3
**Originality:** 4
**Overall Recommendation:** 4
**Confidence:** 3

**Summary:**

The paper uses interpretability techniques to investigate specific failure modes of vision-language models in multi-object visual tasks, such as confusing which features are associated with which objects.

First, using three open-weight VLMs (Qwen, InternVL, Gemma), concept vectors are learned based on a synthetic dataset, either using a probe-based- or centroid-based approach. The validity of the learned vectors is verified by using them for steering model responses (e.g. describing an object as "blue" instead of "red" when steering towards blue). Similarity profiles of different colors are computed based on these vectors and the paper shows that in visual search and similarity tasks, similarities of color representations correlate with errors in two tasks: (i) Given a synthetic image with objects of different shapes and colors, ask if there is an object with a specific color and shape, and (ii) Show an image with labeled colored squares and query image with another colored square, asking the model to output the label corresponding to the color shown in the query image.

The findings suggest that feature representations (color in this case) overlap and the resulting inference in the presence of multiple objects are causally related to certain failure modes.

**Compliance With Llm Reviewing Policy:**

Affirmed.

**Final Justification:**

The rebuttal addressed my main concerns regarding soundness, which I'm increasing from 2 to 3. With the new information provided during the rebuttal, the work seems solid overall. The strengths of original experiments and interesting findings now outweigh the initial issues regarding presentation and I am increasing my score from 3 to 4.

**Key Questions For Authors:**

Color is a rather difficult concept (at least in natural images) but is the feature most centrally used in the paper: Why did the authors choose to focus on color (other than simplicity of generating synthetic images)? Is there any related work on how color is represented in VLMs or other computer vision models?

**Limitations:**

yes

**Strengths And Weaknesses:**

The paper is generally well-written, includes an original set of experiments and provides valuable insights into fundamental limitations of VLMs. However, there are some limitations such as missing information and questionable choices around probing methodology, which seem important to address before publishing.

## Soundness

The experiments are well-chosen and provide a rather comprehensive view on the relationship between feature representation and two failure modes in multi-object tasks.

However, there are some shortcomings which raise questions about the validity of some conclusions:

- For probe-based methods, the paper only uses a single probe architecture they call "attention probes" which is not commonly used in previous literature.
  - There is insufficient information to know whether the probes are actually performing well (also see my points under "Presentation").
  - There are many other probe architectures (e.g., mean aggregation, using the last token, softmax) so it is unclear whether the inadequacy of probe-based vectors is merely due to the chosen architecture and training dataset.
- Steering with probes does not use a global scaling factor as is common in previous literature, but scales based on natural feature activations (as also mentioned in the paper). While this is an interesting approach, the standard approach should also be tried, especially given that steering with probes turns out to perform very poorly.
- Most experiments are done on abstract synthetic images. Transferability is only checked for the color feature based on 60 natural images. As transferability to natural images is critical, it would be great to check this for the shape feature as well.
- It is somewhat misleading to talk about comparing different methods (probe-based vs centroid-based) for extracting concept vectors, but then using different datasets to train these methods (see Figure 1). Why not train a probe on the dataset used for the centroid-based approach?
  - Relatedly, refering to probes as supervised approach in contrast to the centroid-based approach is somewhat confusing (e.g., Section 3 intro and 3.2). Centroids also use concept labels (and are in fact very similar to the difference-of-means approach which is commonly used as baseline in probe papers). Simply refering to "probe-based" vs "centroid-based" seems perfectly fine and less confusing.


## Presentation

The paper is mostly very well-written and the figures are visually appealing, but some important information on probing methodology and datasets is missing.

- Probing methods:
  - The end of Section 3.2 (last half of PCA part) is rather hard to follow and does not explain how the $2N-2$ principal components are assigned to concepts.
  - It is not explained how the probes are trained. In particular, was a train-test split used, how were hyperparameters optimized (if at all), what were performances on the test split? (Some of this can be figured out based on the code repository but at least basic details should be provided in the paper.)
- The appendix contains a list of steps for constructing training datasets, but I could not find some basic information such as:
  - How many samples do the datasets contain?
  - How were training and test splits generated?
- In Sections 4.3-4.5, it is not entirely clear which of the three methods was used to generate the concept vectors. It might have been stated elsewhere (and suppose it is the centroid-based method) but I highly recommend stating the method in these sections.


## Significance

The paper provides insights into a fundamental limitation of VLMs related to superposition, which seems important. While the selection of concepts is restricted to simple geometric shapes and colors, and experiments with natural images are limited, it can inspire other researchers to extend the work to more diverse settings.


## Originality

The chosen experiments are simple yet sensible and provide a rather holistic view. Relating the results to findings from cognitive science is also very interesting (generalization gradients in Section 4.3, and the binding problem).

## Minor Points (Supplementary Materials)

I appreciate that source code is provided for reproducibility, which includes scripts for generating the synthetic datasets, but

- There are minor formatting issues in the code, which prevents it from running out-of-the-box
- Please make sure to include dependencies when publishing your repository, as there currently is no dependencies file but various dependencies are used (numpy, torch etc.)

---

> ### Author Rebuttal · Authors · 2026-03-31
>
> Thanks for the constructive feedback. We address the key question first, then each weakness.
>
> **Why Color?** Our choice is theoretically motivated; color is the canonical feature in the binding problem literature: Treisman & Gelade (1980) showed illusory conjunctions through color-shape misbindings. Also, color is a continuous physical dimension (hue) that VLMs compress into discrete linguistic categories, generating the semantic warping in § 4.3. This connects to Shepard's (1958) generalization gradients, Berlin & Kay's (1969) color universals, and Zaslavsky et al.'s (2018) efficient compression.
> Abdou et al. (CoNLL 2021) further showed that even ungrounded language models develop perceptual-like color representations. We extends this by showing that in VLMs, the visual encoder's continuous representation is further warped by the LLM's discrete token space. By contrast, shape is inherently categorical, lacking both this continuous-to-discrete tension and the fine-grained similarity profiles of § 4.3.
>
> **Alternative Probe Architectures.**
> The attention probe was selected for its highest probing accuracy (>98%, see answer to 3wFb).
> Following your suggestion, we added mean aggregation probes (test accuracy: 88.3% / 73.2% / 88.8% for Qwen/Intern/Gemma.
> Steering results:
>
> | |Qwen|Intern|Gemma|
> |---|:-:|:-:|:-:|
> |Probe|0.0%|2.0%|0.0%|
> |PCA Probe|44.2%|46.8%|17.9%|
> |Centroid|78.1%|35.9%|75.3%|
> |Avg Probe|53.5%|2.9%|10.4%|
> |PCA Avg Probe|37.8%|3.0%|20.8%|
>
> Avg probes underperform on discrimination yet yield better steering, though never matching the top method.
> PCA regularization is inconsistent. All methods nonetheless exhibit the same compositional structure (Appendix C), reinforcing our main claim: color–shape conjunctions are linearly and compositionally represented across models.
> Last-token probes are less motivated since the vision tower lacks causal masking, so the final token carries no privileged summary.
>
> **Steering Scaling Factor.**
> We originally opted for natural feature activation scaling because it minimizes disruption to unrelated visual information.
> However, per your suggestion, we tested the standard global scaling on Qwen2.5-7B probes, alongside our original activation-dependent scaling. Both yield 0% combined steering success. This supports our interpretation that the core limitation is the geometry/quality of probe-derived directions, rather than the specific scaling rule.
>
> **PCA Clarification.** We will revise § 3.2 to make the procedure more explicit. PCA is applied to the full set of $N^2$ probe-derived vectors, and we retain only the first $2N{-}2$ components (the degrees of freedom of the two categorical axes). Each vector is projected as $\mathbf{v}_{PCA}=VV^\top\hat{\mathbf{v}}$, preserving dominant shared modes of variation and removing residual directions that may reflect label memorization, acting as a geometric regularizer toward functionally relevant conceptual axes.
>
> **Fair Comparison: Probes Trained on Centroid Data.**
> We agree that using different datasets for the two methods introduces a potential confound. Our original choice was architecture-driven: probes need positive/negative contrast, while centroids need uncontaminated per-concept activations.
> To rule out this confound, we trained probes on the same single-object data as the centroid.
> On Qwen, these probes yield 0.0% steering success, matching the original multi-object probes and in stark contrast with centroid's 78.1%, consistently across all three models.
> In particular, even these failing probes recover the compositional block structure visible in the centroid-based similarity matrices (Appendix C, Fig. 9).
> The probes correctly identify the compositional subspace, but the specific directions they find are not the ones the model causally uses.
> This dissociation between observational geometry and causal relevance is itself a key finding, motivating steering-based validation as the gold standard for concept vector extraction.
>
> **Other issues:**
> - Regarding transferability to natural images for shapes: see response to reviewer 3AEA, where we report shape steering results on natural images from the VQA dataset.
> - We will explicitly state the method used at the beginning of each experimental section. § 4.3–4.5 use centroid-based vectors throughout.
> - Regarding terminology, we agree with your assessment: centroids also use concept labels, making the "supervised" distinction confusing. We will update the terminology to strictly use "probe-based" and "centroid-based" methods.
> - Training details: ~1100 balanced images per probe, independent test set generated with the same procedure, SGD with lr 10^{-3}, early stopping on training loss, test accuracy >98%.
> - Code: we have fixed the formatting issues that prevented out-of-the-box execution, and added a dependency file. We thank the reviewer for testing the code and apologize for the inconvenience.

---

> > ### Author Rebuttal · Reviewer_b3fs · 2026-04-02
> >
> > Thank you for the comprehensive response. This resolved my concerns and I will increase my score.
> >
> > I encourage the authors to include these new experimental results on mean aggregation probes and probes trained on the single-object dataset into the paper, as these results help to get across the interesting point that observational geometry and causal relevance can correspond to different directions.

---

> > > ### Author Response · Authors · 2026-04-08
> > >
> > > We sincerely thank the reviewer for the thorough engagement with our rebuttal.
> > > We are glad our responses addressed the main concerns, and we will incorporate all the discussed improvements in the revision.

---

### Official Review · Reviewer_3AEA · 2026-03-12

**Soundness:** 2
**Presentation:** 3
**Significance:** 3
**Originality:** 3
**Overall Recommendation:** 4
**Confidence:** 4

**Summary:**

This paper studies why vision language models fail on simple multi object tasks. The authors think the reason is the geometry of the internal representations. They extract concept vectors that represent visual concepts such as color or object type from several open weight VLMs like Qwen, InternVL, and Gemma. They then use steering experiments to check if these vectors really control the model behavior. For example they can push the model to see a red flower as blue. The paper finds that when different concept vectors overlap more in the representation space, the model is more likely to make errors such as mixing color and shape or hallucinating objects. The authors argue that these failures come from interference between concepts in the shared latent space.

**Compliance With Llm Reviewing Policy:**

Affirmed.

**Final Justification:**

Raised my confidence score from 3 to 4.

**Key Questions For Authors:**

Have the authors tried evaluating this framework on existing VLM benchmarks such as visual reasoning or VQA tasks? It would help show whether the geometric interference also explains failures in realistic benchmarks.


The concept vectors are extracted using synthetic datasets. How sensitive are the results to the choice of training data? Would the conclusions remain the same if concept vectors were extracted from natural images?


Since the concept vectors can steer model perception, have the authors explored using them to correct model predictions during inference?


The experiments focus on simple attributes like color and shape. Do the authors expect the same geometric structure to appear for more complex concepts?

**Limitations:**

The experiments focus on simple visual attributes such as color and shape. It is unclear whether the same conclusions hold for more complex visual concepts or real world scenes.


Many experiments use synthetic data. Future work could evaluate the framework on more realistic datasets and tasks.


The work mainly provides analysis of model failures. It does not yet show how to use these insights to improve model performance or robustness.

**Strengths And Weaknesses:**

**Strengths**:

The motivation is interesting and important. Vision language models often fail in multi object scenes, such as mixing color and shape or hallucinating objects. This paper studies the reason behind these failures.

The paper compares different methods to extract concept vectors. The experiments clearly show that some methods work much better than others. This comparison is useful for future work on mechanistic interpretability.

The experimental section contains many interesting observations. For example, the paper shows that overlap between concept vectors can predict model errors in visual search tasks and similarity tasks. This provides useful insight into how internal representations relate to model behavior.

**Weaknesses**:

Many experiments rely on synthetic datasets and simple visual concepts such as colors and shapes. These settings are somewhat toy. It would be more convincing to test the method on more realistic benchmarks or real world tasks.


In practice, modern VLMs often perform reasonably well on natural images. The failures highlighted in this work mainly appear in carefully designed settings. It is therefore unclear how large the impact of this problem is in real applications.


The work mainly focuses on analysis and understanding. It does not show how the proposed findings can improve model performance. For example, it would be interesting to use these concept vectors during inference to reduce errors.

---

> ### Author Rebuttal · Authors · 2026-03-31
>
> We appreciate the Reviewer's detailed and stimulating feedback; we address their questions below.
>
> **Existing Benchmarks**
> We agree that expanding our predictions to well-established benchmarks could be a useful development; however, it would require localizing representations for arbitrary objects from natural scenes, a fairly open problem in mechanistic interpretability. Moreover, we do not claim geometric interference explains all failures on general vision benchmarks, which often involve spatial reasoning, grounding, and other mechanisms well beyond the scope of this work. We also note that Campbell et al. (2024), which we cite, specifically studies binding failures in VLMs using controlled behavioral experiments; our work provides a mechanistic explanation for the failures they documented. Studying how far interference alone can account for errors on other benchmarks is a natural and interesting direction for future work.
>
> **Transferability from Synthetic to Natural Images:**
> To further address this concern, we include a preliminary shape steering experiment on natural images from the VQA/COCO Dataset. We filter for questions containing “What shape is” whose answer matches one of our six probed shapes  (297 examples; distribution: square 126, circle 75, triangle 47, heart 28, star 20, pentagon 1).
> As in the color-transfer experiment, we steer only examples the model answers correctly (183/177/111 for Qwen/Intern/Gemma), applying steering once per alternative shape.
>
> **Steering of Shapes on Natural Images - success rate**
>
> | |Qwen|Intern|Gemma|
> |--|:-:|:-:|:-:|
> |Probe|0.3%|2.6%|0.2%|
> |PCA Probe|9.3%|22%|0.9%|
> |Centroid|19.7%|1.1%|25.8%|
> |Avg Probe|15.6%|0.2%|18.4%|
>
> Results are weaker than color steering but still meaningful: Qwen and Gemma achieve 19.7% and 25.8% with centroids, while InternVL reaches 22% with PCA probes - same pattern of Table 2. These numbers should be read against the strict evaluation criterion, which requires all three sub-tasks (removal, insertion, control) to succeed simultaneously.
>
> The weaker shape transfer reflects a fundamental asymmetry between color and shape as visual attributes - one that also motivates our focus on color (see Why Color? in b3fs). Color is a physical property intrinsic to the object's surface, largely context-independent, and continuously grounded in perception. This is why concept vectors extracted from synthetic stimuli transfer robustly to natural images: the synthetic and natural color distributions share the same underlying perceptual structure.
>
> Shape, by contrast, is inherently categorical and more subtle; there may be a semantic gap between "being a square" (an object identity) and "having a squared shape" (a geometric property): a pool can be "round," a road sign "triangular," a cloud "heart-shaped". Synthetic shapes - uniform, isolated, unambiguous - are too trivial relative to natural images, and the extracted directions do not capture the richer, more distributed representations that natural shape concepts require. Consistently, unlike color steering, shape steering depends strongly on both source and target shape identity (detailed tables below).
>
> We note that capturing richer concepts is not a limitation in principle: with more complex synthetic datasets, richer concept vectors can be extracted. Our goal here is not to provide a general-purpose concept extraction pipeline, but to demonstrate how the geometry of internal representations causally drives model errors - a goal that color, with its continuous perceptual grounding, serves particularly well.
>
> **Centroid Concept Vectors - natural shape steering accuracy**
>
> |Original Shape|qwen|intern|gemma|
> |---|--:|--:|--:|
> |square|1%|0%|2%|
> |triangle|3%|0%|42%|
> |circle|1%|1%|5%|
> |pentagon|-|-|40%|
> |star|86%|4%|65%|
> |heart|68%|3%|41%|
>
> |End Shape|qwen|intern|gemma|
> |---|--:|--:|--:|
> |square|36%|1%|35%|
> |triangle|28%|3%|27%|
> |circle|25%|1%|39%|
> |pentagon|11%|0%|13%|
> |star|14%|1%|15%|
> |heart|13%|1%|31%|
>
>
> **Steering to correct model predictions**
> Using concept vectors to improve inference-time performance is an appealing direction, but our results suggest it is non-trivial. Our finding that high-similarity distractors predict errors (§4.4) implies that naive amplification of the "correct" concept vector would simultaneously amplify representations of confusable objects.
> Designing corrections that navigate this geometry is a substantive research problem that our analysis now makes well-posed, but it is beyond the scope of this paper.
>
> We believe that identifying and characterizing the geometric mechanisms underlying VLM failures is a necessary precursor to proposing corrections.
>
> **Extension to complex properties**
> We have no reason to believe this geometric structure to be specific to color or shape; these were chosen for their controllability and well-defined ground truth, which enabled rigorous validation. Extending the framework to more complex concepts is a natural direction for future work.

---

> > ### Author Rebuttal · Reviewer_3AEA · 2026-04-03
> >
> > Thanks for the detailed and honest response, I understand that the author left many critical questions to future work. I will raise my confidence and keep my score.

---

> > > ### Author Response · Authors · 2026-04-08
> > >
> > > We thank the reviewer for the engaging discussion; we are glad the rebuttal helped clarify the scope and intent of the paper. As the reviewer noted, our contribution is primarily one of analysis and understanding: we characterize the geometric mechanisms underlying VLM failures in multi-object tasks, offering a mechanistic explanation for phenomena that have so far been observed only behaviorally. We believe that precisely identifying and formalizing these failure modes is a necessary first step before proposing principled corrections, and we are confident that this work can serve as a useful foundation to stimulate the community toward exploring intervention approaches.

---

### Official Review · Reviewer_3wFb · 2026-03-14

**Soundness:** 3
**Presentation:** 2
**Significance:** 3
**Originality:** 4
**Overall Recommendation:** 4
**Confidence:** 3

**Summary:**

This paper analyzes visual representations (i.e. output embeddings of the vision encoder) in vision-language models to understand why these models fail at multi-object tasks. This paper claims that these failures are a result of "geometric representational interference", where representations of multiple objects in a scene are not sufficiently disentangled.

The authors attempt to extract feature vectors corresponding to certain concepts using (1) linear probing, (2) the centroid of embeddings corresponding to the concept, and (3) by probing for concept combinations and applying PCA. The authors validate the causal significance of each method's features using activation steering. They conduct a set of systematic experiments using these concept vectors with Qwen, InternVL, and Gemma 3.

**Compliance With Llm Reviewing Policy:**

Affirmed.

**Final Justification:**

I think this paper investigates an important problem, presents original findings, and uses reasonable methods. I think the paper needs improvements to presentation and writing (as discussed). Thanks to the authors for their engagement with my questions and concerns. I would like to maintain my score of Weak Accept and I think the textual improvements we discussed will be very helpful to the paper.

**Key Questions For Authors:**

- I raised some concerns above regarding the high-level objectives of this paper (as described) and how these are precisely addressed by each experiment. Can the authors please restate these more directly (with clarity and precision)?

- Since visual encodings are projected into LLM input space, would using the vocabulary embedding for a particular concept also make sense?

- L156 mentions an issue that probes may overfit to their training distribution. Can you report the training set accuracy, so we can determine the degree of overfitting?

**Limitations:**

Yes

**Strengths And Weaknesses:**

**Soundness:** The methods and experiment details look reasonable to me. My concerns are about the high-level claims of this paper. First, I understand from the introduction that the goal is to better understand failures in "multi-object tasks". I think only Sec. 4.4 and 4.5 concern multi-object scenes, and these are very particular tasks. I think the introduction needs to be much more precise with the claims about which problems are caused by the limitations in "representational interference". Research question 1 (L036) asks about "fundamental architectural constraints" (and again in L410). I don't agree with this framing, because this paper only engages with pre-trained models. I think theory or controlled empirical studies (e.g. training toy models on synthetic data) are required to evaluate "fundamental" capabilities.

**Presentation:** I think the paper's structure could definitely be clearer. The immediate goals and experimental designs of this paper are quite hard to understand from the introduction. The connection between representational geometry and "multi-object tasks" is imprecise, the analogies to other systems would be better placed in discussion, and core claims are not intuitive (why is it that "structured and compositional representations introduce susceptibility to interference between concepts"?). I think Section 4 would benefit a lot from a roadmap. This reads like a loosely motivated sequence of experiments --- it is quite hard to predict what will come next and understand what the overall takeaways are.

**Significance:** The problems this paper is explaining (e.g. the binding problem in vision–language models) are of great interest and understanding how representations influence model behavior is important and under-explored.

**Originality:** I understand that the findings of this work are new. And their experimental design and tasks appear to be tailored to this work and are also original.

---

> ### Author Rebuttal · Authors · 2026-03-31
>
> We thank the Reviewer for their constructive feedback and address each point below.
>
> **High-level objectives and presentation.**
> Our paper makes three linked claims, each addressed by specific experiments:
> - *VLMs develop compositional and semantically structured representations.* Color-shape concept vectors exhibit clean factorization (§ 4.2), and continuous color representations form a semantically warped manifold aligned with linguistic categories (§ 4.3).
> - *These representations are causally active.* Steering confirms that the extracted directions functionally control model perception on both synthetic and natural images (§ 4.1–4.2). Only geometrically faithful directions (centroids) succeed; discriminatively learned probes fail despite high classification accuracy.
> - *Representational geometry predicts downstream failures.* Cosine similarity between concept vectors correlates with accuracy degradation in visual search (§ 4.4) and governs model confidence in color similarity (§ 4.5), establishing a direct link between representational geometry and task-level errors.
>
> We agree that this logical progression is not sufficiently explicit in the current manuscript.
> In the revision, we will:
> (i) state these three claims as a roadmap at the start of § 4;
> (ii) tighten the introduction to state the experimental goals; and
> (iii) move the extended cognitive analogies into a dedicated discussion section, where they can be developed without disrupting the empirical narrative.
>
> **Theoretical ground for central claims**
> As detailed above, our central argument is that compositional, semantically organized representations introduce geometric similarity between distinct object encodings, and that this similarity is the proximate cause of binding failures. The connection is direct: if a "red square" is represented as the sum of a "red" and a "square" component, then "red square" and "red circle" necessarily share a component, placing them closer in latent space than objects sharing no feature. In a single-object setting this supports generalization; in multi-object scenes it becomes a liability, as the model must resolve feature bindings from partially overlapping representations. This is precisely what § 4.4 and 4.5 demonstrate empirically.
> We will rephrase Research Question 1 as: *"Do these errors reflect systematic constraints arising from the architectural design and training paradigm of current VLMs, or artifacts of specific model conditions?"* Our argument is not that VLMs are incapable of these tasks, but that the training of this architecture on broad next-token prediction encourages factored, semantically organized representations — and this same structure introduces interference. This is consistent with Shai et al. (arXiv:2602.02385, 2026), who show a bias towards factored representations in transformers, and Nurisso et al. (2025), who formalize this tradeoff theoretically and validate it in toy networks; we extend their framework by identifying, in pre-trained VLMs, representations exhibiting precisely the predicted properties. The convergence of representational geometry across three architecturally distinct models (RSA $r>0.93$, Table 3) indicates these are systematic constraints rather than model-specific artifacts - we will accordingly revise "fundamental architectural constraints" to "shared representational constraints." Our work complements Campbell et al. (2024), who document the behavioral signatures, and Assouel et al. (ICLR 2026), who identify the responsible circuits; we characterize the representational geometry that determines when those circuits fail. This three-level picture (theory, circuits, geometry) is consistent and mutually reinforcing.
>
> **Vocabulary Embeddings as Concept Vectors.**
> Vocabulary embeddings reside in the same space as our concept vectors and are valid candidates in principle, but encode linguistic semantics rather than the visual-perceptual structure we study - bypassing the compression through the visual encoder that drives the failures we document. A direct comparison would nonetheless be informative: alignment would confirm the semantic warping in §4.3; divergence would indicate a qualitatively different visual encoding.
>
> **Probe Accuracy and Discriminative Shortcuts.**
> Train/test accuracy (below) shows a negligible gap, ruling out classical overfitting. However, the point raised in L156 is subtler: probes can generalize perfectly on the discrimination task while still learning directions that are not causally relevant. Our steering results demonstrate exactly this: near-perfect test coexists with near-zero steering success. The discriminative objective admits many separating hyperplanes, and the one found need not coincide with the direction the model actually uses.
>
> Attention Probes Accuracy
>
> |  |Qwen|Intern|Gemma|
> |---|:---:|:---:|:---:|
> |Train Set|99.9%|98.0%|99.0%|
> |Test Set|99.9%|98.0%|98.7%|
>
> New results\&discussion will be included in the revised manuscript.

---

> > ### Author Rebuttal · Reviewer_3wFb · 2026-04-04
> >
> > Thanks to the authors for answering my questions. I am satisfied with their response and I think most importantly it will be pertinent to see the improvements to the presentation of their paper that they have promised.

---

> > > ### Author Response · Authors · 2026-04-08
> > >
> > > We thank the reviewer for the thoughtful engagement throughout this discussion. We appreciate the reviewer's recognition of the work's originality and significance, and we are grateful for the reviewer's precise and constructive criticism regarding the paper's structure and framing. We are confident that the presentation improvements we have committed to - the roadmap at the start of Section 4, the tightened introduction, the revised framing of our central claims, and the relocation of the extended cognitive analogies to a dedicated discussion section - will make the paper's logical progression substantially clearer and more accessible to readers.

---

### Decision · Program_Chairs · 2026-04-30

**Decision:**

Accept (regular)

**Comment:**

Reviewers agree that this is a solid paper which provides a significant and original contribution. Concerns regarding soundness were adequately addressed during the rebuttal. Minor concerns regarding clarity remain, but which can hopefully be addressed for the final version of the paper.
I therefore recommend accepting this paper to ICML.